# The Scabbard of Excalibur: An Allegory on the Role of an Efficient and Effective Healthcare System under Universal Health Coverage during the Pandemic Response

**DOI:** 10.3390/healthcare12100979

**Published:** 2024-05-09

**Authors:** Hiroyuki Noda

**Affiliations:** 1Public Health, Department of Social Medicine, Graduate School of Medicine, Osaka University, Suita 565-0871, Japan; hiroyuki-noda@umin.net; Tel.: +81-6-6879-3911; 2Public Health Bureau, Ministry of Health, Labour and Welfare, Tokyo 100-8916, Japan

**Keywords:** COVID-19 response, universal health coverage, vaccination, healthcare system efficiency, pandemic preparedness, allegorical analysis

## Abstract

During the COVID-19 pandemic, while some countries succeeded in reducing their rate of death after SARS-CoV-2 infection via vaccination by the end of 2021, some of them also faced hospital capacity strain, leading to social anxiety about delays in the diagnosis and treatment of patients with other diseases. This essay presents an allegory to explain the situation during the COVID-19 pandemic. Through an allegory and *Le Morte d’Arthur (Arthur’s Death)*, this essay indicates that “the scabbard of Excalibur” that we are looking for is an efficient and effective healthcare system that can diagnose patients who might become severely ill due to COVID-19 and to treat them without hospital capacity strain. In *Le Morte d’Arthur*, the scabbard of Excalibur was lost, and we have not been able to find any alternatives to end the COVID-19 pandemic. We can choose a future in which “the scabbard of Excalibur” exists, providing a different ending for the next pandemic.

## 1. Introduction

In the early 2020s, the pandemic of 2019’s coronavirus disease, which was named coronavirus disease 2019 (COVID-19) by the World Health Organization (WHO) on 11 February 2020 [1], was among the main public health challenges around the world [2]. On 5 May 2023, the Director-General of the WHO declared that COVID-19 was an established and ongoing health issue that no longer constituted a public health emergency of international concern (PHEIC) [3]. On the other hand, there is no rigorous quantitative definition of pandemics, let alone their endings [4,5,6], and the statement during the fifteenth meeting of the IHR (2005) Emergency Committee on the COVID-19 pandemic held on 4 May 2023 indicated that “it is time to transition to long-term management of the COVID-19 pandemic” [3].

The COVID-19 pandemic caused worldwide turbulence affecting health [7,8,9], the economy [10,11], social lives [12], and the politics of nations [13,14,15]. Although vaccination reduced the rate of death after SARS-CoV-2 infection [16,17], the healthcare system sometimes became overwhelmed by a huge number of patients with mild to moderate illnesses [18,19]. It was difficult to end the COVID-19 pandemic by promoting vaccination alone, but we have not been able to find any alternatives other than the declaration of the end of the PHEIC by the Director-General of the WHO as a “transition to long-term management of the COVID-19 pandemic” [3] or the declarations ending the national emergencies in each country [20].

In times of crisis, many of us are strongly drawn to history [21]. During the COVID-19 pandemic, we turned to it for a desperately needed perspective and ideally for useful lessons from past pandemics and epidemics, including the 1918 pandemic influenza and the Black Death [22,23,24]. During future pandemics, many may turn to our perspective of COVID-19 and other pandemics or epidemics in the hope to see it through our lenses. Therefore, how people recall the COVID-19 pandemic may be crucial in future societal debates on pandemic preparedness and appropriate political action [25,26].

This essay demonstrates the importance of an efficient and effective healthcare system under universal health coverage against pandemics, in addition to vaccination, through an allegory of the COVID-19 pandemic.

## 2. Methods

As preparation for this essay, narrative reviews were conducted to examine the real-world effect of vaccination on deaths after SARS-CoV-2 infections, hospital strain during the COVID-19 pandemic, and the use of lessons from past pandemics and epidemics during the COVID-19 pandemic via citation search (i.e., a snowball search) [27] starting with PubMed. Through these narrative reviews, 6 articles on the real-world effect of vaccination on deaths after SARS-CoV-2 infections, 16 articles on hospital strain during the COVID-19 pandemic, and 14 articles on the use of lessons from past pandemics and epidemics during the COVID-19 pandemic were identified.

In this allegory, metaphors are used to explain the situation during the COVID-19 pandemic (Table 1). Epidemiological terminology and concepts, including SARS-CoV-2 (i.e., the three-headed dragon), changes in variants (i.e., a spell of magic), the effect of COVID-19 on deaths (i.e., fire of death), the effect of COVID-19 on illness (i.e., fire of illness), the effect of COVID-19 on SARS-CoV-2 infections (i.e., fire of infection), the vaccine (i.e., Excalibur), and booster vaccination (i.e., the third slash), are used as sources of metaphors in this allegory. Previous studies have suggested that vaccination reduced the rate of deaths after SARS-CoV-2 infection, as well as the rate of severe cases, in some of the highly vaccinated countries by the end of 2021 [28,29,30,31], but the number of patients with mild to moderate illness due to COVID-19 and the number of infected people have not significantly decreased in these countries [32,33]. A previous study indicated that vaccination may reduce the risk of SARS-CoV-2 infection in individual-level analyses, yet vaccination also provided at least modest levels of individual protection [34].

Because Excalibur is the legendary sword in the Arthurian legend, this essay cites text from *Le Morte d’Arthur (Arthur’s Death)*. The cited text of *Le Morte d’Arthur* is the original text edited from the Winchester Manuscript and Caxton’s *Morte d’Arthur*, which was edited by P.J.C. Field. It was first published in 2017 and reprinted in 2019 [35]. The story of the *Morte d’Arthur* forms a great arc, starting with Arthur’s birth and ending with his death, and the central tales focus on Arthur’s knights as the protagonists of those adventures [35]. The scabbard of Excalibur, as well as Excalibur, is also used as a metaphor in this essay. Because the scabbard of Excalibur had powers of its own, preventing the wearer from ever bleeding to death in battle, this essay uses the scabbard of Excalibur as a metaphor for an efficient and effective healthcare systems under universal health coverage.

## 3. An Allegory: Excalibur and the Three-Headed Dragon

Once upon a time, there was a big dragon with three heads. The three-headed dragon suddenly appeared in the Far East, and it instantly terrorized the world. In some countries, they sacrificed freedom and human rights to seal the dragon. In other countries, they tried to live with the dragon, resulting in a large number of deaths.

The dragon had three heads and the ability to breathe a different type of fire from each mouth. The first was a head that breathed “fire of death”, which caused death if not treated, leading to an increment in deaths among the public. The second was a head that breathed “fire of illness”, which increased the number of patients with mild to moderate illnesses. They rarely died, leading to hospital capacity strain and a fear of death among the public. The third was a head that breathed “fire of infection”, which absorbed people as nutrition, strengthening the dragon. After more than a year of fighting, several million people had died, and people had been terrorized around the world.

About a year later, a knight with Excalibur appeared. Receiving the wishes of the public, the knight stood in front of the three-headed dragon with Excalibur in his hand. In the blink of an eye, the knight cut off the first head by slashing twice. By cutting off the head that breathed “fire of death”, the number of deaths decreased dramatically among the public, but the fires of the two remaining heads continued to burn people.

To resist the knight’s next slash, the dragon cast the magic spell “Delta” to harden its scales. On the other side, the knight raised Excalibur over his head. They glared at one another in silence.

The silence echoed upon the air.

In the next moment, the knight slashed the dragon, aiming to cut off the two remaining heads. The third slash tore the hardened scales, but the dragon endured it. The dragon squeezed the force while casting another magic spell, and from the remaining heads, it breathed more fire toward the knight than ever before…

## 4. The Road to the Battle of Camlann

This story presents the situation during the COVID-19 pandemic as an allegory of SARS-CoV-2 (i.e., the three-headed dragon) and the vaccine (i.e., Excalibur) against it. By promoting vaccination, we succeeded in reducing deaths after SARS-CoV-2 infections as well as severe cases in some of the highly vaccinated countries by the end of 2021 [28,29,30,31], but the number of patients with mild to moderate illness due to COVID-19 and the number of infected people have not significantly decreased in these countries [32,33]. On the contrary, they are increasing, mainly due to the Delta variant (i.e., the magic of “Delta”) and the subsequent Omicron variant (i.e., the other magic), whereas mutations can act as a double-edged sword for any virus, including SARS-CoV-2, where genetic alterations can generate either attenuated or strengthened viruses [36,37]. Some countries have faced hospital capacity strain due to the huge numbers of patients with mild to moderate illnesses due to COVID-19, leading to social anxiety about delays in the diagnosis and treatment of patients with other diseases [38,39,40].

In the Arthurian legend, Excalibur was the legendary sword of King Arthur and had magical powers. In *Le Morte d’Arthur* [35], King Arthur, via Merlin, obtained Excalibur from the Lady of the Lake but finally commanded that Excalibur be cast into the water after he was fatally wounded in the final battle. The allegory above is an unfinished story about the COVID-19 pandemic, but the Arthurian legend ends with the Battle of Camlann, which is known as the Battle of Salisbury in *Le Morte d’Arthur*.

As the destination of the allegory about the COVID-19 pandemic, two types of endings are expected. One is an ending where the knight succeeds in cutting off all the heads with Excalibur, and the other is an ending where the knight cannot cut off the rest of the heads. We will have a happy ending if we proceed to the former, but it is doubtful whether this will ever be achieved. That is, it is difficult to reach herd immunity [41], and the Delta variant (i.e., the magic of “Delta”) and others (e.g., the Omicron variant) reduced the effect of vaccination in preventing SARS-CoV-2 infections [42,43,44,45,46]. The booster vaccination (i.e., the third slash) may have improved the effect in preventing SARS-CoV-2 infections [47], but there continues to be no evidence that herd immunity has been reached in community settings. King Arthur could use magical powers with Excalibur, but he was fatally wounded in the final battle. Like the Arthurian legend, we may be on the road to the Battle of Camlann (or the Battle of Salisbury) in the COVID-19 pandemic.

If we proceed to the latter ending, we will need an additional item other than Excalibur—the scabbard of Excalibur—to protect ourselves from the “fire of illness” that the second head breathes out.

## 5. In Search of the Scabbard of Excalibur


*Than seyde Merlion, “Whethir lyke ye bettir, the swerde othir the scawberde?”*



*“I lyke bettir the swerde,” seyde Arthure.*



*“Ye ar the more unwyse, for the scawberde ys worth ten of the swerde; for whyles ye have the scawberde uppon you ye shall lose no blood, be ye never so sore wounded. Therefore kepe well the scawberde allweyes with you.”*
*Le Morte d’Arthur* by Sir Thomas Malory in 1485 [35].

The scabbard of Excalibur also had powers of its own, preventing the wearer from ever bleeding to death in battle. Merlin, who was known as a prophet as well as a magician, pointed out that the scabbard was more valuable than Excalibur, and he advised King Arthur to keep it on his person all the time. However, the scabbard of Excalibur was stolen from King Arthur and lost in the Arthurian legend, leading to his death after the final battle.

In the allegory, as well as in the COVID-19 pandemic, it is difficult to live with the three-headed dragon (i.e., SARS-CoV-2) due to the increase in deaths caused by the fire of the first head (i.e., “fire of death”) and the overflow of patients caused by the fire of the second head (i.e., “fire of illness”). If only the third head (i.e., the head breathing “fire of infection”) remains, there will only be infected patients without a fear of death, which will make it possible to live with the dragon, as with the common cold.

On the other hand, if the second head remains, hospitals will continue to be overwhelmed with each wave of COVID-19 infection to save patients who could die among the huge numbers of patients with mild to moderate illnesses. “The scabbard of Excalibur” that we are looking for is a healthcare system that can efficiently diagnose patients who might become severely ill among the huge numbers of patients with mild to moderate illness, and one that can effectively treat them without hospital capacity strain under universal health coverage.

After a long fight against SARS-CoV-2, we have developed predictive models of mortality risk and severity using readily available clinical and laboratory data [48], and we give therapeutic antibodies to patients with mild to moderate illness in clinical settings [49]. In addition, some pharmaceutical companies have developed oral antivirals [50,51,52,53] for non-hospitalized patients with a high risk of severe illness as an easy-to-administer treatment. Furthermore, portable testing systems for SARS-CoV-2 with small battery-driven devices have been used to detect COVID-19 patients [54,55]. Strategic use of these technologies may help us to establish efficient and effective healthcare systems under universal health coverage for protection from the “fire of illness” in the case of the remaining second head of the pandemic or endemic.

## 6. Universal Health Coverage and the Scabbard of Excalibur

Universal health coverage was the important global goal, even before the COVID-19 pandemic began [56,57], and it is also key in dealing with health crises, as illustrated during the COVID-19 pandemic [58,59]. Historically, the member states of the WHO committed in 2005 to developing their health financing systems so that all people have access to services and do not suffer financial hardship paying for them [56], and this goal was defined as universal coverage, sometimes called universal health coverage [60]. According to the WHO, “universal health coverage (UHC) means that all people have access to the full range of quality health services they need, when and where they need them, without financial hardship” [61], and its access has three dimensions: physical accessibility, financial affordability, and acceptability [60]. Physical accessibility is understood as the availability of good health services within reasonable reach of those who need them and of opening hours, appointment systems, and other aspects of service organizations and delivery that allow people to obtain services when they need them. Financial affordability is a measure of people’s ability to pay for services without financial hardship, and acceptability is people’s willingness to seek services [60,62]. Moreover, services must be physically accessible, financially affordable, and acceptable to patients if universal health coverage is to be attained [60].

During the COVID-19 pandemic, some countries such as Japan achieved financial affordability by waiving medical care costs relating to COVID-19 [63,64], and since it was common for people to visit clinics and hospitals for the diagnosis and treatment of diseases and illnesses even before the COVID-19 pandemic, acceptability was also achieved according to the perspective of the three dimensions. On the other hand, achieving physical accessibility was difficult in many countries, including developed countries with higher values on the UHC effective coverage index [65] and lower values for excess mortality during the COVID-19 pandemic [7,8,9]. For example, a report by Kassenärztliche Bundesvereinigung in Germany suggests that the fact that the vast majority of COVID-19 patients have been treated on an outpatient basis since the beginning of the pandemic has made it possible to prevent the dreaded overload of inpatient structures. This report indicates that doctors’ offices act as the first “Schutzwall” (i.e., defensive wall), allowing hospitals to concentrate on caring for serious cases [66]. Moreover, a report by experts meeting with the Government of Japan suggests that it took time to set up a system to provide medical care for people recuperating at home and in other facilities, as well as outpatient clinics for fever patients, and concludes that it will be important to go even further and develop a system that will allow primary care physicians to exercise their functions [67]. Although it may be difficult to increase the number of health professionals during a pandemic, efficient use of existing healthcare systems may be a better way to fight against pandemics. These reports support the importance of healthcare system efficiency under universal health coverage for protection against hospital strain to prepare for the next pandemic.

On the other hand, the situation may differ depending on the implementation levels of universal health coverage as well as pandemic preparedness. Previous studies suggested that universal health coverage was a key concept in the COVID-19 response [58,59] and that universal health coverage mitigated the health-related consequences of COVID-19 in Asia and Oceania [58]. However, a previous study showed differences in outcomes regarding COVID-19 between continents and also within the continent of Africa and attempted to explain these differences using some potential factors such as the seeding effect, testing capacity, and population [68]. Historically, each country has developed its own national health system to suit its own circumstances, and each country will develop its own unique approach to national health systems [69]. It was pointed out that systems of medical care are based on the history of the medical care and lifestyle habits of the people in each country [70]. Therefore, we may have to find solutions that suit the systems in each country as “the scabbard of Excalibur”.

We have not been able to find any alternatives to end the COVID-19 pandemic other than the declaration of the end of the PHEIC by the Director-General of the WHO [3] or the declarations ending the national emergencies in each country [20], but these declarations themselves are nothing more than policy procedures. It may be a better public health policy to end the pandemic using “the scabbard of Excalibur” of efficient and effective healthcare systems under universal health coverage.

## 7. Conclusions

*Le Morte d’Arthur* ended after the scabbard of Excalibur was lost, and we have not been able to find any alternatives to end the COVID-19 pandemic other than the declaration of the end of the PHEIC by the Director-General of the WHO or the declarations ending the national emergencies in each country. However, we can choose a future in which “the scabbard of Excalibur” exists for the next pandemic. In fact, some countries have begun healthcare reform to prepare for the next pandemic. We may have to find solutions that suit the systems in each country as “the scabbard of Excalibur”, but attaining efficient and effective healthcare systems under universal health coverage, as the scabbard of Excalibur, will help us prepare for the next pandemic.

## Figures and Tables

**Table 1 healthcare-12-00979-t001:** Metaphors regarding epidemiological terminology and concepts in this essay.

Epidemiological Terminology and Concepts	Metaphors in This Essay
SARS-CoV-2	Three-headed dragon
Changes in variants	Spell of magic
Effect of COVID-19 on deaths	Fire of death
Effect of COVID-19 on illness	Fire of illness
Effect of COVID-19 on SARS-CoV-2 infections	Fire of infection
Vaccine	Excalibur
The booster vaccination	The third slash
Efficient and effective healthcare systems under universal health coverage	The scabbard of Excalibur

## Data Availability

Not applicable.

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
