# Peer review of "The Scabbard of Excalibur: An Allegory on the Role of an Efficient and Effective Healthcare System under Universal Health Coverage during the Pandemic Response"

_healthcare, 2024, doi:10.3390/healthcare12100979_

Round 1
Reviewer 1 Report
Comments and Suggestions for Authors
Below are some comments to consider:
· The imaginative allegory of Excalibur and the three-headed dragon could be strengthened by connecting it to real-world data and scientific evidence. It may be good to directly link each dragon "head" to public health metrics or outcomes validated by recent research or data.
· The UHC discussion is crucial, but it might be expanded to include how different countries' health policies affected pandemic response. Comparative analysis with nations with different UHC implementation levels may reveal its effectiveness amid a global health crisis.
· It would improve the text to briefly describe or cite the methodology behind forecasts or models, such as those for severe COVID-19 instances. This can make the findings transparent.
· Verify all references are current and appropriate. Historical pandemic references could be updated or augmented with more recent literature to reflect the newest research.
· The manuscript may utilize figurative terminology like "fire of death" or "fire of illness," which may confuse readers expecting a scholarly debate. Clarify these terms or use epidemiological terminology.
· The manuscript needs proofreading to correct minor grammatical errors and improve sentence structure for readability.
· Adding figures or tables summarizing key points, such as the impact of vaccination on severe cases, can enhance reader engagement and comprehension.
Comments on the Quality of English LanguageMinor editing of English language required
Author Response
FOR REVIEWER1
Comments and Suggestions for Authors
Below are some comments to consider:
>> Thank you very much for valuable comments. I revised my manuscript point by point as follows.
The imaginative allegory of Excalibur and the three-headed dragon could be strengthened by connecting it to real-world data and scientific evidence. It may be good to directly link each dragon "head" to public health metrics or outcomes validated by recent research or data.
>> Thank you very much for valuable comments. To “directly link each dragon "head" to public health metrics or outcomes”, I added table (P2) and the descriptions as follows; “In this allegory, metaphors are used to explain the situation during the COVID-19 pandemic (Table). Epidemiological terminology and concepts, including SARS-CoV-2 (i.e., the three-headed dragon), changes in variants (i.e., a spell of magic), the effect of COVID-19 on deaths (i.e., fire of death), the effect of COVID-19 on illness (i.e., fire of illness), the effect of COVID-19 on SARS-CoV-2 infections (i.e., fire of infection), the vaccine (i.e., Excalibur), and booster vaccination (i.e., the third slash), are used as sources of metaphors in this allegory. Previous studies suggest that vaccination reduced the rate of deaths after SARS-CoV-2 infection, as well as the rate of severe cases, in some of the highly vaccinated countries by the end of 2021 [28-31], but the number of patients with mild to moderate illness due to COVID-19 and the number of infected people have not significantly decreased in these countries [32,33]. A previous study indicated that vaccination may reduce the risk of SARS-CoV-2 infection in individual-level analyses, but it provided at least modest levels of individual protection [34].” (P2L62).
- The UHC discussion is crucial, but it might be expanded to include how different countries' health policies affected pandemic response. Comparative analysis with nations with different UHC implementation levels may reveal its effectiveness amid a global health crisis.
>> I added and revised the descriptions as follows; “On the other hand, the situation may differ depending on the implementation levels of universal health coverage as well as pandemic preparedness. Previous studies suggested that universal health coverage was a key concept in the COVID-19 response [58,59] and that universal health coverage mitigated the health-related consequences of COVID-19 in Asia and Oceania [58]. However, a previous study showed differences in outcomes regarding COVID-19 between continents and within the continent of Africa and attempted to explain their reasons using some potential factors such as the seeding effect, testing capacity, and population [68]. Historically, each country has developed its own national health system to suit its own circumstances, and each country will develop its own unique approach to national health systems [69]. It was pointed out that systems of medical care are based on the history of the medical care and lifestyle habits of the people in each country [70]. Therefore, we may have to find solutions that suit the systems in each country as “the scabbard of Excalibur”.” (P5L228).
- It would improve the text to briefly describe or cite the methodology behind forecasts or models, such as those for severe COVID-19 instances. This can make the findings transparent.
>> To briefly describe or cite the methodology behind forecasts or models, I added table (P2) and the descriptions as follows; “In this allegory, metaphors are used to explain the situation during the COVID-19 pandemic (Table). Epidemiological terminology and concepts, including SARS-CoV-2 (i.e., the three-headed dragon), changes in variants (i.e., a spell of magic), the effect of COVID-19 on deaths (i.e., fire of death), the effect of COVID-19 on illness (i.e., fire of illness), the effect of COVID-19 on SARS-CoV-2 infections (i.e., fire of infection), the vaccine (i.e., Excalibur), and booster vaccination (i.e., the third slash), are used as sources of metaphors in this allegory. Previous studies suggest that vaccination reduced the rate of deaths after SARS-CoV-2 infection, as well as the rate of severe cases, in some of the highly vaccinated countries by the end of 2021 [28-31], but the number of patients with mild to moderate illness due to COVID-19 and the number of infected people have not significantly decreased in these countries [32,33]. A previous study indicated that vaccination may reduce the risk of SARS-CoV-2 infection in individual-level analyses, but it provided at least modest levels of individual protection [34]. Because Excalibur is the legendary sword in the Arthurian legend, this essay cites text from Le Morte d’Arthur (Arthur's Death). The cited text of Le Morte d’Arthur is the original text edited from the Winchester Manuscript and Caxton’s Morte d’Arthur, which was edited by P.J.C. Field. It was first published in 2017 and reprinted in 2019 [35]. The story of the Morte d’Arthur forms a great arc, starting with Arthur’s birth and ending with his death, and the central tales focus on Arthur’s knights as the protago-nists of those adventures [35]. The scabbard of Excalibur, as well as Excalibur, is also used as a metaphor in this essay. Because the scabbard of Excalibur had powers of its own, preventing the wearer from ever bleeding to death in battle, this essay uses the scabbard of Excalibur as a metaphor for an efficient and effective healthcare systems under universal health coverage.” (P2L62).
- Verify all references are current and appropriate. Historical pandemic references could be updated or augmented with more recent literature to reflect the newest research.
>> I updated “historical pandemic references” and added reference number 23 and 24 as follows (P7L311).
- Doran, Á.; Colvin, C.L.; McLaughlin, E. What can we learn from historical pandemics? A systematic review of the lit-erature. Soc Sci Med 2024, 342, 116534.
- Jin, SL.; Kolis, J.; Parker, J.; Proctor, D.A.; Prybylski, D.; Wardle, C.; Abad, N.; Brookmeyer, K.A.; Voegeli, C.; Chiou, H. Social histories of public health misinformation and infodemics: case studies of four pandemics. Lancet Infect Dis 2024 (Online ahead of print).
- The manuscript may utilize figurative terminology like "fire of death" or "fire of illness," which may confuse readers expecting a scholarly debate. Clarify these terms or use epidemiological terminology.
>> To clarify these terms using epidemiological terminology.”, I added table (P2) and descriptions as follows; “In this allegory, metaphors are used to explain the situation during the COVID-19 pandemic (table). Epidemiological terminology and concepts, including SARS-CoV-2 (i.e., the three-headed dragon), changes in variants (i.e., a spell of magic), the effect of COVID-19 on deaths (i.e., fire of death), the effect of COVID-19 on illness (i.e., fire of illness), the effect of COVID-19 on SARS-CoV-2 infections (i.e., fire of infection), the vaccine (i.e., Excalibur), and booster vaccination (i.e., the third slash), are used as sources of metaphors in this allegory.” (P2L62).
- The manuscript needs proofreading to correct minor grammatical errors and improve sentence structure for readability.
>> I revised the manuscript using English editing serves recommended by MDPI.
- Adding figures or tables summarizing key points, such as the impact of vaccination on severe cases, can enhance reader engagement and comprehension.
>> To summarize the key points in this essay and the allegory, I added table (P2). I also added the descriptions as follows; “In this allegory, metaphors are used to explain the situation during the COVID-19 pandemic (Table). Epidemiological terminology and concepts, including SARS-CoV-2 (i.e., the three-headed dragon), changes in variants (i.e., a spell of magic), the effect of COVID-19 on deaths (i.e., fire of death), the effect of COVID-19 on illness (i.e., fire of illness), the effect of COVID-19 on SARS-CoV-2 infections (i.e., fire of infection), the vaccine (i.e., Excalibur), and booster vaccination (i.e., the third slash), are used as sources of metaphors in this allegory. Previous studies suggest that vaccination reduced the rate of deaths after SARS-CoV-2 infection, as well as the rate of severe cases, in some of the highly vaccinated countries by the end of 2021 [28-31], but the number of patients with mild to moderate illness due to COVID-19 and the number of infected people have not significantly decreased in these countries [32,33]. A previous study indicated that vaccination may reduce the risk of SARS-CoV-2 infection in individual-level analyses, but it provided at least modest levels of individual protection [34].” (P2L62).
Reviewer 2 Report
Comments and Suggestions for Authors
Dear Author,
Thank you for your submission. Your essay seems to cover a relevant topic, as COVID-19 pandemic was a public health issue with unprecedented impact. It is important to explore perspectives on how to control it in the future. Please read my contribution to improve your essay.
TITLE AND ABSTRACT
Consider the following title: "The Scabbard of Excalibur: An Allegory on the Role of Efficient Healthcare Systems and Universal Coverage in Pandemic Response."
Lines 10 to 13: You can improve the comparison or contrast you presented. As I understood, you want to convey the idea that vaccination was effective in some countries but not so much in others. However, the way you wrote, it can be confusing for some readers because you say, "[some countries] have succeeded in reducing the death rate (...) by vaccination (...) but some (...) faced hospital capacity strain (...).
A country can reduce the death rate and still face hospital capacity strain. Also, hospital strain is more directly associated with prevalence, not necessarily mortality. In my opinion, you can improve the argument by either focusing in vaccination vs. prevalence/hospital strain or vaccination vs. mortality alone.
Line 16: remove "among a huge number of the patients with mild to moderate illness.". I think the text is unnecessary.
Line 20: Regarding the keywords, consider "COVID-19 response" instead of "COVID-19" pandemic to be more specific. Furthermore, consider adding "healthcare system efficiency," "pandemic preparedness," and "alegorical analysis.".
INTRODUCTION
Line 23: instead of "coronavirus disease 2019" write "2019's coronavirus disease".
Line 25: remove "now"
Lines 27 and 28: remove ", and the declaration came more than three years after the outbreak of COVID-19 started [1]"
Line 29: change "their end" to "its ending."
Line 30: I do not know if I understood. Was the meeting held in 2005? By that time, the pandemic was not known. Please review.
Lines 33 to 36: Overcitation: Review the number of citations. For instance, in line 34, you cite sources 6 to 18 (12 sources) to support a simple sentence. If the same applies to the citations "[19–24]" (6 sources) and [25–40] (15 sources),. You did the same in lines 45 [24–55] (13 sources) and 153 [86–96] (10 sources).
It gives the false impression that many sources were necessary to write some parts of your essay. When some topic is broad, search for a review article or documents with multidisciplinary contents and you do not have to cite a large number of documents.
Line 46: change "epidemic and pandemic, including the COVID-19 pandemic," to "COVID-19 and other pandemics or epidemics."
Line 47: eliminate "that we will give them perspective on the present by allowing them."
Line 47–48: change "the lenses of our perspectives" to "our lens."
Line 48: instead of "SARS-CoV-2," say "COVID-19.". The same virus can cause future pandemics but COVID-19 is from 2019.
Lines 51 to 53: You must mention that it is through an allegory with the book Le Morte d'Arthur.
NOTE
There must be a brief explanation on your method. Where did you get the book? Which edition? Present the book in the references. Being a classic, it is not difficult to find. Which sections of the book are relevant? All of it or specific ones?
Then, explain about the articles you used to critically review your topic. Where and how did you find them? The internet? Through which search engine? What were the key words? What were the inclusion and exclusion criteria? How many documents suitable for this discussion did you find?
Then, how did you perform your analysis, i.e., how did you compare the story with the sources you found?
Also, explain step-by-step how you prepared your discussion by basically explaining what the sections of your essay are.
SECTION 2
At some point of your essay (maybe a separate section), you can draw a table presenting the parallelisms between the elements of your story and the pandemic. For example:
Dragon: SARS-CoV-2
Dragon heads: SARS-CoV-2 impacts (is it?)
Excalibur, a vaccine
Spells: SARS-CoV-2 variants
etc.
Lines 55 to 58: There is also the case of African countries, where the dragon seemed comparatively harmless, perhaps because they had been struggling with other dragons (https://doi.org/10.1016/j.jnma.2020.10.001). It is worth mentioning, even if it is not the main focus of this essay.
Lines 94–99: You should have told the story of King Arthur before telling the allegory to facilitate understanding among readers unfamiliar with it. It could even be in the introduction because it is the basis of your allegory.
SECTION 4 (in search of Excalibur)
Besides all the issues mentioned, section 4 presents a solid discussion. I think most of the essay could follow the same rhetoric.
SECTION 4 (should be 5) - UHC
Lines 152-169: should be part of the introduction. Important concepts should all be introduced at the beginning to facilitate the readers' comprehension.
NOTE: Your essay is a critical review using an allegory. It should follow the structure Overview (relevant facts) -> Introduction (key concepts, problem, rationale, and objectives) -> Research methodology (as I explained above) -> Main scenarios to choose and why each -> Thesis statement, clarifying your choice or perspective (in your case, it could be UHC) -> Arguments in favor -> Drawbacks or limitations -> Discussion (why the arguments in favor outweigh the drawbacks) -> Conclusion -> Recommendations -> Contingency plan if UHC turns impractical
CONCLUSION
The conclusion proposes that each country should find its own approach towards UHC, considering its reality. This would provide some preparedness for future pandemics of COVID or other diseases.
It is an understandable conclusion, although it raises further questions, perhaps to be discussed in other essays or studies. Still, you could provide some insights on how countries with different realities could implement UHC to control COVID-19. Furthermore, you mentioned that the end of PHEIC did not necessarily mean the end of the pandemic. Did WHO make the best decision? In the light of Arthur's allegory, what should and shall WHO do? I also feel that your discussion on vaccination could have been more robust. Perhaps your essay could be less about vaccination and more about UHC as a whole (choose the focus between vaccination or UHC).
IN SUMMARY
Proofread and restructure your essay according to the recommendations above and bring
REFERENCES
Reference "[1]" indicates a document from 2005. Is that correct? There are many journal articles on the topic (find the link at the end of my comments).
There are some broken links in your reference list. Please make sure that all urls are functional.
https://scholar.google.com/scholar?hl=pt-PT&as_sdt=0%2C5&q=allintitle%3A+%22COVID-19%22+%22PHEIC%22&btnG=
Yours sincerely
Comments on the Quality of English LanguageDear author,
Regarding grammar, I recommend a thorough English writing review, perhaps with a native or advanced speaker. Meanwhile, please pay attention to the following:
Line 11: it feels redundant to say "vaccination in some of the highly vaccinated". Choose different wording without repeating the idea of vaccine.
Lines 10 to 12: you can improve the writing. The text "we have succeeded (...) but some of them" does not flow very well, though I understood it. Instead, you can say, some "some countries suceeded (...) while others..."
Line 13: the book's title is "Le Morte d'Arthur" (Arthur's Death).
Line 15: replace "severe illness" with "severely ill"
Line 17: In the end of "Le Morte d'Arthur", the scabbard of Excalibur was lost...
Line 23: instead of "had been" write "was".
Line 24: instead of "one of the crucial" write "among the main".
Line 95: I am unfamiliar with the word "gat." Is that what you wanted to write? And please review the grammar in lines 94 to 99.
Line 139: change "severe illness" to "severely ill"
Yours sincerely
Author Response
FOR REVIEWER2
Dear Author,
Thank you for your submission. Your essay seems to cover a relevant topic, as COVID-19 pandemic was a public health issue with unprecedented impact. It is important to explore perspectives on how to control it in the future. Please read my contribution to improve your essay.
>> Thank you very much for valuable comments. I revised my manuscript point by point as follows.
TITLE AND ABSTRACT
Consider the following title: "The Scabbard of Excalibur: An Allegory on the Role of Efficient Healthcare Systems and Universal Coverage in Pandemic Response."
>> I changed title as follows; “The Scabbard of Excalibur: An Allegory on the Role of an Efficient and Effective Healthcare System under Universal Health Coverage during the Pandemic Response” (P1L2).
Lines 10 to 13: You can improve the comparison or contrast you presented. As I understood, you want to convey the idea that vaccination was effective in some countries but not so much in others. However, the way you wrote, it can be confusing for some readers because you say, "[some countries] have succeeded in reducing the death rate (...) by vaccination (...) but some (...) faced hospital capacity strain (...).
A country can reduce the death rate and still face hospital capacity strain. Also, hospital strain is more directly associated with prevalence, not necessarily mortality. In my opinion, you can improve the argument by either focusing in vaccination vs. prevalence/hospital strain or vaccination vs. mortality alone.
>> I agree with your suggestions. To focus on vaccination vs. hospital strain, I changed the descriptions as follows; “while some countries succeeded in reducing their rate of death after SARS-CoV-2 infection via vaccination by the end of 2021, some of them also faced hospital capacity strain” (P1L10).
Line 16: remove "among a huge number of the patients with mild to moderate illness.". I think the text is unnecessary.
>>I removed the description. (P1L16).
Line 20: Regarding the keywords, consider "COVID-19 response" instead of "COVID-19" pandemic to be more specific. Furthermore, consider adding "healthcare system efficiency," "pandemic preparedness," and "alegorical analysis.".
>> I changed the keywords as follows; “the COVID-19 response; universal health coverage; vaccination; healthcare system efficiency; pandemic preparedness; allegorical analysis” (P1L20).
INTRODUCTION
Line 23: instead of "coronavirus disease 2019" write "2019's coronavirus disease".
>> I changed the description as follows; “In the early 2020s, the pandemic of 2019's coronavirus disease” (P1L24). As WHO originally named this disease as “coronavirus disease 2019”, I added the description as follows; “which was named coronavirus disease 2019 (COVID-19) by the World Health Organization (WHO) on February 11th, 2020 [1]” (P1L24).
Line 25: remove "now"
>>I removed “now”. (P1L27).
Lines 27 and 28: remove ", and the declaration came more than three years after the outbreak of COVID-19 started [1]"
>>I removed the description. (P1L29).
Line 29: change "their end" to "its ending."
>> I changed the description into “their endings” (P1L30), based on the English editing service recommended by MDPI.
Line 30: I do not know if I understood. Was the meeting held in 2005? By that time, the pandemic was not known. Please review.
>> Thank you very much for valuable comments. This meeting held in 2023, but the name of this meeting is “The International Health Regulations (2005) Emergency Committee regarding the outbreak of novel coronavirus (2019-nCoV)”. I checked the descriptions in WEB page of WHO, and this is correct. The word ”2005” indicate the year in which the IHR was amended, and the current IHR is named as “The International Health Regulations (2005)”, so this conference is named as “The International Health Regulations (2005) Emergency Committee regarding the outbreak of novel coronavirus (2019-nCoV)”.
Lines 33 to 36: Overcitation: Review the number of citations. For instance, in line 34, you cite sources 6 to 18 (12 sources) to support a simple sentence. If the same applies to the citations "[19–24]" (6 sources) and [25–40] (15 sources),. You did the same in lines 45 [24–55] (13 sources) and 153 [86–96] (10 sources).
It gives the false impression that many sources were necessary to write some parts of your essay. When some topic is broad, search for a review article or documents with multidisciplinary contents and you do not have to cite a large number of documents.
>> I reduced the references as follows; “health [7-9], the economy [10,11], social lives [12], and the politics of nations [13-15]” (P1L34), “[18,19]” (P1L38),“[22-24]” (P2L46),“[38-40]” (P3L129), and“[56,57]” (P5L190) as relevant cited references.
Line 46: change "epidemic and pandemic, including the COVID-19 pandemic," to "COVID-19 and other pandemics or epidemics."
>> I changed the description as follows; “COVID-19 and other pandemics or epidemics” (P2L47).
Line 47: eliminate "that we will give them perspective on the present by allowing them."
>>I removed the description. (P2L47).
Line 47–48: change "the lenses of our perspectives" to "our lens."
>> I changed the description into “our lens” (P2L47).
Line 48: instead of "SARS-CoV-2," say "COVID-19.". The same virus can cause future pandemics but COVID-19 is from 2019.
>> I changed the description as follows; “COVID-19” (P2L48).
Lines 51 to 53: You must mention that it is through an allegory with the book Le Morte d'Arthur.
NOTE
There must be a brief explanation on your method. Where did you get the book? Which edition? Present the book in the references. Being a classic, it is not difficult to find. Which sections of the book are relevant? All of it or specific ones?
Then, explain about the articles you used to critically review your topic. Where and how did you find them? The internet? Through which search engine? What were the key words? What were the inclusion and exclusion criteria? How many documents suitable for this discussion did you find?
Then, how did you perform your analysis, i.e., how did you compare the story with the sources you found?
Also, explain step-by-step how you prepared your discussion by basically explaining what the sections of your essay are.
>> I added the brief explanation regarding “the book Le Morte d'Arthur” used in this essay as well as citation search as follows; “Because Excalibur is the legendary sword in the Arthurian legend, this essay cites text from Le Morte d’Arthur (Arthur's Death). The cited text of Le Morte d’Arthur is the original text edited from the Winchester Manuscript and Caxton’s Morte d’Arthur, which was edited by P.J.C. Field. It was first published in 2017 and reprinted in 2019 [35]. The story of the Morte d’Arthur forms a great arc, starting with Arthur’s birth and ending with his death, and the central tales focus on Arthur’s knights as the protagonists of those adventures [35]. The scabbard of Excalibur, as well as Excalibur, is also used as a metaphor in this essay. Because the scabbard of Excalibur had powers of its own, preventing the wearer from ever bleeding to death in battle, this essay uses the scabbard of Excalibur as a metaphor for an efficient and effective healthcare systems under universal health coverage.” (P2L79) and “As preparation for this essay, narrative reviews were conducted to examine the real-world effect of vaccination on deaths after SARS-CoV-2 infections, hospital strain during the COVID-19 pandemic, and the use of lessons from past pandemics and epi-demics during the COVID-19 pandemic via citation search (snowball search) [27] starting with PubMed. Through these narrative reviews, 6 articles on the real-world effect of vaccination on deaths after SARS-CoV-2 infections, 16 articles on hospital strain during the COVID-19 pandemic, and 14 articles on the use of lessons from past pan-demics and epidemics during the COVID-19 pandemic were identified.” (P2L54).
SECTION 2
At some point of your essay (maybe a separate section), you can draw a table presenting the parallelisms between the elements of your story and the pandemic. For example:
Dragon: SARS-CoV-2
Dragon heads: SARS-CoV-2 impacts (is it?)
Excalibur, a vaccine
Spells: SARS-CoV-2 variants
etc.
>> I added table (P2).
Lines 55 to 58: There is also the case of African countries, where the dragon seemed comparatively harmless, perhaps because they had been struggling with other dragons (https://doi.org/10.1016/j.jnma.2020.10.001). It is worth mentioning, even if it is not the main focus of this essay.
>> Thank you very much for valuable comments. I added the description as follows; “a previous study showed differences in outcomes regarding COVID-19 between continents and within the continent of Africa and attempted to explain their reasons using some potential factors such as the seeding effect, testing capacity, and population [68].” (P5L232).
Lines 94–99: You should have told the story of King Arthur before telling the allegory to facilitate understanding among readers unfamiliar with it. It could even be in the introduction because it is the basis of your allegory.
>> I added the description as follows; “The story of the Morte d’Arthur forms a great arc, starting with Arthur’s birth and ending with his death, and the central tales focus on Arthur’s knights as the protagonists of those adventures [35].” (P2L82).
SECTION 4 (in search of Excalibur)
Besides all the issues mentioned, section 4 presents a solid discussion. I think most of the essay could follow the same rhetoric.
>> Thank you very much for valuable comments. Although the descriptions regarding solid discussion were increased due to other reviewers’ comments, I added and revised the descriptions with same rhetoric as follows; “Historically, each country has developed its own national health system to suit its own circumstances, and each country will develop its own unique approach to national health systems [69]. It was pointed out that systems of medical care are based on the history of the medical care and lifestyle habits of the people in each country [70]. Therefore, we may have to find solutions that suit the systems in each country as “the scabbard of Excalibur”. We have not been able to find any alternatives to end the COVID-19 pandemic other than the declaration of the end of the PHEIC by the Director-General of the WHO [3] or the declarations ending the national emergencies in each country [20], but these declarations themselves are nothing more than policy procedures. It may be a better public health pol-icy to end the pandemic using “the scabbard of Excalibur” of efficient and effective healthcare systems under universal health coverage.” (P5L235). I also added method section to explain Metaphors in this essay as follows; “In this allegory, metaphors are used to explain the situation during the COVID-19 pandemic (Table). Epidemiological terminology and concepts, including SARS-CoV-2 (i.e., the three-headed dragon), changes in variants (i.e., a spell of magic), the effect of COVID-19 on deaths (i.e., fire of death), the effect of COVID-19 on illness (i.e., fire of illness), the effect of COVID-19 on SARS-CoV-2 infections (i.e., fire of infection), the vaccine (i.e., Excalibur), and booster vaccination (i.e., the third slash), are used as sources of metaphors in this allegory. Previous studies suggest that vaccination reduced the rate of deaths after SARS-CoV-2 infection, as well as the rate of severe cases, in some of the highly vaccinated countries by the end of 2021 [28-31], but the number of patients with mild to moderate illness due to COVID-19 and the number of infected people have not significantly decreased in these countries [32,33]. A previous study indicated that vaccination may reduce the risk of SARS-CoV-2 infection in individual-level analyses, but it provided at least modest levels of individual protection [34]. Because Excalibur is the legendary sword in the Arthurian legend, this essay cites text from Le Morte d’Arthur (Arthur's Death). The cited text of Le Morte d’Arthur is the original text edited from the Winchester Manuscript and Caxton’s Morte d’Arthur, which was edited by P.J.C. Field. It was first published in 2017 and reprinted in 2019 [35]. The story of the Morte d’Arthur forms a great arc, starting with Arthur’s birth and ending with his death, and the central tales focus on Arthur’s knights as the protagonists of those adventures [35]. The scabbard of Excalibur, as well as Excalibur, is also used as a metaphor in this essay. Because the scabbard of Excalibur had powers of its own, preventing the wearer from ever bleeding to death in battle, this essay uses the scabbard of Excalibur as a metaphor for an efficient and effective healthcare systems under universal health coverage.” (P2L62)
SECTION 4 (should be 5) - UHC
Lines 152-169: should be part of the introduction. Important concepts should all be introduced at the beginning to facilitate the readers' comprehension.
NOTE: Your essay is a critical review using an allegory. It should follow the structure Overview (relevant facts) -> Introduction (key concepts, problem, rationale, and objectives) -> Research methodology (as I explained above) -> Main scenarios to choose and why each -> Thesis statement, clarifying your choice or perspective (in your case, it could be UHC) -> Arguments in favor -> Drawbacks or limitations -> Discussion (why the arguments in favor outweigh the drawbacks) -> Conclusion -> Recommendations -> Contingency plan if UHC turns impractical
>> Thank you very much for valuable comments. I added the description as method section as follows; “In an allegory, metaphor was used to explain situation during the COVID-19 pan-demic (Table). The epidemiological terminologies and concepts including SARS-CoV-2 (i.e., three-headed dragon), change of variant (i.e., a spell of magic), effect of COVID-19 for deaths (i.e., fire of death), effect of CIVID-19 for illness (i.e., fire of illness), effect of COVID-19 for SARS-CoV-2 infections (i.e., fire of infection), vaccine (i.e., Excalibur), and booster vaccination (i.e., the third slash) were used as metaphors in this allegory. Previous studies suggest that vaccination reduced deaths after SARS-CoV-2 infections as well as severe cases in some of the highly vaccinated countries as of the end of 2021 [58-61], but the number of patients with mild to moderate illness by COVID-19 and the infected people have not significantly decreased in these countries [62-64]. Because Excalibur is the legendary sword in Arthurian legend, this essay cited a text of Le Morte Darthur. The cited text of Le Morte Darthur was the original text edited from the Winchester Manuscript and Caxton’s Morte Darthur, which was edited by P.J.C. Field, firstly published in 2017 and reprinted in 2019 [70]. The story of the Morte Darthur forms a great arc, starting with Arthur’s birth and ending with his death, and the central tales fo-cus on Arthur’s knights as the protagonists of those adventures [70]. The Scabbard of Excalibur as well as Excalibur is also used as metaphor in this essay. Because the Scabbard of Excalibur had powers of its own, preventing the wearer from ever bleeding to death in battle, this essay used the Scabbard of Excalibur as a metaphor of Universal Health Coverage.” (P2L62), and separate and revised the section 4 (i.e., section 5 and section 6).
CONCLUSION
The conclusion proposes that each country should find its own approach towards UHC, considering its reality. This would provide some preparedness for future pandemics of COVID or other diseases.
It is an understandable conclusion, although it raises further questions, perhaps to be discussed in other essays or studies. Still, you could provide some insights on how countries with different realities could implement UHC to control COVID-19. Furthermore, you mentioned that the end of PHEIC did not necessarily mean the end of the pandemic. Did WHO make the best decision? In the light of Arthur's allegory, what should and shall WHO do? I also feel that your discussion on vaccination could have been more robust. Perhaps your essay could be less about vaccination and more about UHC as a whole (choose the focus between vaccination or UHC).
>> Thank you very much for valuable comments. I added the descriptions as follows; “On the other hand, the situation may differ depending on the implementation levels of universal health coverage as well as pandemic preparedness. Previous studies suggested that universal health coverage was a key concept in the COVID-19 response [58,59] and that universal health coverage mitigated the health-related consequences of COVID-19 in Asia and Oceania [58]. However, a previous study showed differences in outcomes regarding COVID-19 between continents and within the continent of Africa and attempted to explain their reasons using some potential factors such as the seeding effect, testing capacity, and population [68]. Historically, each country has developed its own national health system to suit its own circumstances, and each country will develop its own unique approach to national health systems [69]. It was pointed out that systems of medical care are based on the history of the medical care and lifestyle habits of the people in each country [70]. Therefore, we may have to find solutions that suit the systems in each country as “the scabbard of Excalibur”.” (P5L228) and “We have not been able to find any alternatives to end the COVID-19 pandemic other than the declaration of the end of the PHEIC by the Director-General of the WHO [3] or the declarations ending the national emergencies in each country [20], but these declarations themselves are nothing more than policy procedures. It may be a better public health pol-icy to end the pandemic using “the scabbard of Excalibur” of efficient and effective healthcare systems under universal health coverage.” (P6L241). To focus on UHC, I revised the description as follows; “Le Morte d’Arthur ended after the scabbard of Excalibur was lost, and we have not been able to find any alternatives to end the COVID-19 pandemic other than the declara-tion of the end of the PHEIC by the Director-General of the WHO or the declarations end-ing the national emergencies in each country. However, we can choose a future in which “the scabbard of Excalibur” exists for the next pandemic. In fact, some countries are start-ing healthcare reform to prepare for the next pandemic. We may have to find solutions that suit the systems in each country as “the scabbard of Excalibur”, but attaining efficient and effective healthcare systems under universal health coverage, as the scabbard of Ex-calibur, will help us prepare for the next pandemic.” (P6L248).
IN SUMMARY
Proofread and restructure your essay according to the recommendations above and bring
>> I revised the manuscript using English editing serves recommended by MDPI. I revised the abstract as follows; “During the COVID-19 pandemic, while some countries succeeded in reducing their rate of death after SARS-CoV-2 infection via vaccination by the end of 2021, some of them also faced hospital capacity strain, leading to social anxiety about delays in the diagnosis and treatment of patients with other diseases. This essay presents an allegory to explain the situation during the COVID-19 pandemic. Through an allegory and Le Morte d’Arthur (Arthur's Death), this essay indicates that "the scabbard of Excalibur" that we are looking for is an efficient and effective healthcare system to diagnose patients who might become severely ill due to COVID-19 and to treat them without hospital capacity strain. In Le Morte d’Arthur, the scabbard of Excalibur was lost, and we have not been able to find any alternatives to end the COVID-19 pandemic. We can choose a future in which "the scabbard of Excalibur" exists, providing a different ending for the next pandemic.” (P1L10).
REFERENCES
Reference "[1]" indicates a document from 2005. Is that correct? There are many journal articles on the topic (find the link at the end of my comments).
>> Thank you very much for valuable comments. Yes, I checked the descriptions in WEB page of WHO, and this description is correct. The word ”2005” indicate the year in which the IHR was amended, and the current IHR is listed as “The International Health Regulations (2005)”, so this conference regarding the reference number 1 is named “The International Health Regulations (2005) Emergency Committee regarding the outbreak of novel coronavirus (2019-nCoV)”. I checked the description of the reference number 1 (currently the reference number 2) again and described as follows (P6L267).
- The International Health Regulations (2005) Emergency Committee regarding the outbreak of novel coronavirus (2019-nCoV). Statement on the second meeting of the International Health Regulations (2005) Emergency Committee regarding the outbreak of novel coronavirus (2019-nCoV). Available online: https://www.who.int/news/item/30-01-2020-statement-on-the-second-meeting-of-the-international-health-regulations-(2005)-emergency-committee-regarding-the-outbreak-of-novel-coronavirus-(2019-ncov) (accessed on 30 April 2024)
There are some broken links in your reference list. Please make sure that all urls are functional.
https://scholar.google.com/scholar?hl=pt-PT&as_sdt=0%2C5&q=allintitle%3A+%22COVID-19%22+%22PHEIC%22&btnG=
>> Thank you very much for valuable comments. I checked them and described the URLs as follows; “https://www.who.int/docs/default-source/coronaviruse/situation-reports/20200211-sitrep-22-ncov.pdf?sfvrsn=fb6d49b1_2” (P6L265), “https://www.who.int/news/item/30-01-2020-statement-on-the-second-meeting-of-the-international-health-regulations-(2005)-emergency-committee-regarding-the-outbreak-of-novel-coronavirus-(2019-ncov)” (P6L269), “https://www.who.int/news/item/05-05-2023-statement-on-the-fifteenth-meeting-of-the-international-health-regulations-(2005)-emergency-committee-regarding-the-coronavirus-disease-(covid-19)-pandemic” (P6L274), “https://www.whitehouse.gov/wp-content/uploads/2023/01/SAP-H.R.-382-H.J.-Res.-7.pdf” (P7L307), “https://www.who.int/publications/m/item/covid-19-epidemiological-update-15-march-2024” (P7L332), “https://www.who.int/health-topics/universal-health-coverage” (P8L409), “https://www.kbv.de/media/sp/Ambulante_Versorgung_Corona_Pandemie_Zahlen_Fakten.pdf” (P9L420), and “https://www.cas.go.jp/jp/seisaku/coronavirus_yushiki/pdf/corona_kadai.pdf” (P9L423).
Comments on the Quality of English Language
Dear author,
Regarding grammar, I recommend a thorough English writing review, perhaps with a native or advanced speaker. Meanwhile, please pay attention to the following:
>> I revised the manuscript using English editing serves recommended by MDPI.
Line 11: it feels redundant to say "vaccination in some of the highly vaccinated". Choose different wording without repeating the idea of vaccine.
>> I removed the description “in some of the highly vaccinated countries” (P1L11).
Lines 10 to 12: you can improve the writing. The text "we have succeeded (...) but some of them" does not flow very well, though I understood it. Instead, you can say, some "some countries suceeded (...) while others..."
>> I revised the description as follows; “while some countries succeeded in reducing their rate of death after SARS-CoV-2 infection via vaccination by the end of 2021, some of them also faced hospital capacity strain” (P1L10). Because “some of them” are included in “some countries”, I did not use the word “others”.
Line 13: the book's title is "Le Morte d'Arthur" (Arthur's Death).
>> I revised the description as follows; “Le Morte Darthur (Arthur's Death)” (P1L14).
Line 15: replace "severe illness" with "severely ill"
>> I revised the description as follows; “severely ill” (P1L16).
Line 17: In the end of "Le Morte d'Arthur", the scabbard of Excalibur was lost...
>> I revised the description as follows; “In Le Morte d’Arthur, the scabbard of Excalibur was lost,” (P1L17). Because it is not clear when the scabbard of Excalibur was lost in "Le Morte d'Arthur", I did not use the words “the end of”.
Line 23: instead of "had been" write "was".
>> I revised the description as follows; “In the early 2020s, the pandemic of 2019's coronavirus disease, which was named coronavirus disease 2019 (COVID-19) by the World Health Organization (WHO) on February 11th, 2020 [1], was among the main public health challenges around the world [2].” (P1L24).
Line 24: instead of "one of the crucial" write "among the main".
>> I revised the description as follows; “In the early 2020s, the pandemic of 2019's coronavirus disease, which was named coronavirus disease 2019 (COVID-19) by the World Health Organization (WHO) on February 11th, 2020 [1], was among the main public health challenges around the world [2].” (P1L24).
Line 95: I am unfamiliar with the word "gat." Is that what you wanted to write? And please review the grammar in lines 94 to 99.
>> I revised the manuscript using English editing serves recommended by MDPI, and I revised the description as follows; “In the Arthurian legend, Excalibur was the legendary sword of King Arthur and had magical powers. In Le Morte d’Arthur [35], King Arthur, via Merlin, obtained Ex-calibur from the Lady of the Lake but finally commanded that Excalibur be cast into the water after he was fatally wounded in the final battle. The allegory above is an unfin-ished story about the COVID-19 pandemic, but the Arthurian legend ends with the Battle of Camlann, which is known as the Battle of Salisbury in Le Morte d’Arthur.” (P3L130).
Line 139: change "severe illness" to "severely ill"
>> I revised the description as follows; ““The scabbard of Excalibur” that we are looking for is a healthcare system to efficiently diagnose patients who might become severely ill” (P4L173).

Reviewer 3 Report
Comments and Suggestions for Authors
Dear authors
The text reflects on the importance of the strength of national health systems in the fight against the COVID-19 pandemic and against others that may appear. The work is interesting, but it must be reformulated in depth so that it can be published by the journal.
We would like to point out a series of suggestions that the authors should take into account in order for the paper to be published:
-First, although the Arthurian metaphor is appropriate we consider that it should have less weight in the central argument. As a metaphor, once presented, we recommend that the authors use it to deepen the implications it has for the analysis being undertaken.
-Secondly, the length of the text is insufficient, actually the text of the paper occupies four pages. No space sufficient is left for the analysis of why a solid health system is a guarantee in the fight against pandemics such as COVID-19. This analytical-reflexive part, which is central to the work, should be clearly expanded. In its present state it is insufficient.
-The conclusions should be reinforced. We suggest that in them the authors refer to the main milestones achieved with the research.
We encourage the authors to make these changes, which will undoubtedly strengthen their work so that it better meets the requirements for publication.
Author Response
FOR REVIEWER3
Dear authors
The text reflects on the importance of the strength of national health systems in the fight against the COVID-19 pandemic and against others that may appear. The work is interesting, but it must be reformulated in depth so that it can be published by the journal.
We would like to point out a series of suggestions that the authors should take into account in order for the paper to be published:
>> Thank you very much for valuable comments. I revised my manuscript point by point as follows.
-First, although the Arthurian metaphor is appropriate we consider that it should have less weight in the central argument. As a metaphor, once presented, we recommend that the authors use it to deepen the implications it has for the analysis being undertaken.
>> To deepen the implications, I added and revised the description as follows; “For example, a report by Kassenärztliche Bundesvereinigung in Germany suggests that the fact that the vast majority of COVID-19 patients have been treated on an outpatient basis since the beginning of the pandemic has made it possible to prevent the dreaded overload of inpatient structures. This report indicates that doctors’ offices act as the first “Schutzwall” (i.e., defensive wall), allowing hospitals to concentrate on caring for serious cases [66]. Moreover, a report by experts meeting with the Government of Japan suggests that it took time to set up a system to provide medical care for people recu-perating at home and in other facilities, as well as outpatient clinics for fever patients, and concludes that it will be important to go even further and develop a system that will allow primary care physicians to exercise their functions [67]. Although it may be dif-ficult to increase the number of health professionals during a pandemic, but efficient use of existing healthcare systems may be a better way to fight against pandemics. These reports support the importance of healthcare system efficiency under universal health coverage for protection against hospital strain to prepare for the next pandemic.
On the other hand, the situation may differ depending on the implementation levels of universal health coverage as well as pandemic preparedness. Previous studies suggested that universal health coverage was a key concept in the COVID-19 response [58,59] and that universal health coverage mitigated the health-related consequences of COVID-19 in Asia and Oceania [58]. However, a previous study showed differences in outcomes regarding COVID-19 between continents and within the continent of Africa and attempted to explain their reasons using some potential factors such as the seeding effect, testing capacity, and population [68]. Historically, each country has developed its own national health system to suit its own circumstances, and each country will develop its own unique approach to national health systems [69]. It was pointed out that systems of medical care are based on the history of the medical care and lifestyle habits of the people in each country [70]. Therefore, we may have to find solutions that suit the sys-tems in each country as “the scabbard of Excalibur”.” (P5L213).
-Secondly, the length of the text is insufficient, actually the text of the paper occupies four pages. No space sufficient is left for the analysis of why a solid health system is a guarantee in the fight against pandemics such as COVID-19. This analytical-reflexive part, which is central to the work, should be clearly expanded. In its present state it is insufficient.
>> I added and revised the description as follows; “For example, a report by Kassenärztliche Bundesvereinigung in Germany suggests that the fact that the vast majority of COVID-19 patients have been treated on an outpatient basis since the beginning of the pandemic has made it possible to prevent the dreaded overload of inpatient structures. This report indicates that doctors’ offices act as the first “Schutzwall” (i.e., defensive wall), allowing hospitals to concentrate on caring for serious cases [66]. Moreover, a report by experts meeting with the Government of Japan suggests that it took time to set up a system to provide medical care for people recuperating at home and in other facilities, as well as outpatient clinics for fever patients, and concludes that it will be important to go even further and develop a system that will allow primary care physicians to exercise their functions [67]. Although it may be difficult to increase the number of health professionals during a pandemic, but efficient use of existing healthcare systems may be a better way to fight against pandemics. These reports support the importance of healthcare system efficiency under universal health coverage for protection against hospital strain to prepare for the next pandemic.” (P5L213), as there is no limit to the number of words for essay in Healthcare.
-The conclusions should be reinforced. We suggest that in them the authors refer to the main milestones achieved with the research.
>> I added the description as follows; “In fact, some countries are starting healthcare reform to prepare for the next pandemic. We may have to find solutions that suit the systems in each country as “the scabbard of Excalibur”, but attaining efficient and effective healthcare systems under universal health coverage, as the scabbard of Excalibur, will help us prepare for the next pandemic.” (P6L252).
We encourage the authors to make these changes, which will undoubtedly strengthen their work so that it better meets the requirements for publication.
>> Thank you very much for valuable comments.

Round 2
Reviewer 2 Report
Comments and Suggestions for Authors
I accept the current version in the present form.
Reviewer 3 Report
Comments and Suggestions for Authors
The text is now adequated for publishing after the authors have improved it according to the suggestions of reviewer.
Kind regards